# Gender Identification in a Two-Level Hierarchical Speech Emotion Recognition System for an Italian Social Robot

**DOI:** 10.3390/s22051714

**Published:** 2022-02-22

**Authors:** Antonio Guerrieri, Eleonora Braccili, Federica Sgrò, Giulio Nicolò Meldolesi

**Affiliations:** Fondazione Neurone Onlus, Viale Regina Margherita 169, 00198 Roma, Italy; eleonora.braccili@fondazioneneurone.it (E.B.); federica.sgro@fondazioneneurone.it (F.S.); gn.meldolesi@fondazioneneurone.it (G.N.M.)

**Keywords:** Gender Recognition, Speech Emotion Recognition (SER), affective states, Human-Robot Interaction (HRI), social robot

## Abstract

The real challenge in Human-Robot Interaction (HRI) is to build machines capable of perceiving human emotions so that robots can interact with humans in a proper manner. Emotion varies accordingly to many factors, and gender represents one of the most influential ones: an appropriate gender-dependent emotion recognition system is recommended indeed. In this article, we propose a Gender Recognition (GR) module for the gender identification of the speaker, as a preliminary step for the final development of a Speech Emotion Recognition (SER) system. The system was designed to be installed on social robots for hospitalized and living at home patients monitoring. Hence, the importance of reducing the software computational effort of the architecture also minimizing the hardware bulkiness, in order for the system to be suitable for social robots. The algorithm was executed on the Raspberry Pi hardware. For the training, the Italian emotional database EMOVO was used. Results show a GR accuracy value of 97.8%, comparable with the ones found in the literature.

## 1. Introduction

In recent years, researchers, designers, and the general public have been fascinated with the possibility of building able and intelligent machines to engage in social interaction: education, companionship, therapy, and aging-in-place are the most common applications for which social robots have been thought [1]. The pivotal point for socially interactive robots to be successful is the ability to interact with humans in a similar way as humans do, making robots not just a tool but rather collaborators, companions, tutors, and all kinds of social interaction partners [2]. Thus, knowledge about the user’s identity and his/her emotional state must be considered an essential part of the intelligence technology with which social robots have to be equipped to function with sensitivity towards humans. The voice represents the optimal medium for a robot to get both information, a powerful source by which each subject can be uniquely identified and recognized, and the fastest and the most natural way to communicate and express emotions. Hence, the interest in the recognition of emotion by speech.

Speech Emotion Recognition (SER) task is however affected by various factors: recording environment conditions and acoustical acquisition devices are some examples, but above all, emotional expression variability represents the most relevant one. It is well-known from the literature that culture, age, and gender roles have a stronger impact on emotional expression [3,4,5]. In particular, gender information is mainly used by researchers for enhancing SER accuracy due to the gender differences in speech emotional expressiveness [6,7]. A possible explanation of these differences relies on social factors: based on social standards, women are mainly inclined to be more sensitive and calm, unlike men who are considered more impassive and irascible [8].

On the other hand, the physical characteristics of male and female sound generation systems vary according to the anatomical characteristics, such as the length of the vocal tract or the size of the vocal cords; thus, the generated sound (voice) has specific acoustic characteristics (tone, intensity, energy, forming frequencies, etc.) depending on each anatomical structure. Therefore, having a common emotion recognizer model for both sexes may not provide accurate results [9] and a gender-specific SER system is recommended.

In this article, we propose a Gender Recognition (GR) module for the gender identification of the speaker, as a preliminary step for the final development of a GEnder-dependent SPeech Emotion Recognition (GESPER) system. The system was designed to be installed on social robots for hospitalized or living at home patients monitoring. For this reason, the computational effort of the software has been reduced to the minimum in order to make the software executable even on platforms with reduced space and computational capacity, such as in social robots.

## 2. Related Works

Most of the state-of-art SER studies implement a system architecture organized into a two-level recognizer [9,10,11,12] (see Figure 1): first, a gender recognizer predicts the gender of the speaker, and then, depending on the outcome, a gender-specific speech emotion recognizer is used to identify the speaker emotion. Differences in acoustic characteristics for male and female speakers are a well-known problem and it is proven from the literature that gender-specific emotion recognizers improve the overall recognition rate with respect to the gender-independent ones. Accuracy improvement of 5% on average is found in [10,11,12], up to 10.36% in [9].

A different approach is proposed in [4] where instead of considering two different models of emotion, the information retrieved at the gender recognition step is used at a feature-level with a single model, by using a distributed-gender feature approach. Anyway, also, in this case, an accuracy improvement of about 6% was found with respect to the one without the gender distinction.

The most relevant features used for gender recognition are pitch-related: pitch assumes discriminating values usually ranging from 8–180 Hz and 165–255 Hz (in a neutral emotional state) for male and female speakers respectively [13,14] by which the gender is easily recognized. In [11], the authors use an average pitch as a threshold value to classify the gender of the speaker. However, using this method on emotional voices is proved to be less efficient, since vocal men expressions could be confused with those of neutral women, and vice versa. This is a relevant disadvantage for an automatic system, especially compared to human listeners who would easily distinguish the gender of a speaker even in different emotional conditions. A thresholding approach is also used in [15], but additionally a second level identification using GMM has been applied to manage suspicious cases.

Apparently, not only pitch-related features, but also Mel-Frequency Cepstrum Coefficients (MFCC) and energy-related features can play an important role in gender identification. A combination of the above-mentioned features also ensured particularly better results, as demonstrated in [16,17]. Some other approaches were also tried to identify the gender of a speaker directly from raw audio signals like in [18] where a CNN was trained to directly extract features from the raw signal in a filtering stage and then the classification was performed.

All the above-mentioned SER works have been done for the English/German languages. As far as we know, not many efforts have been made at the state-of-art level to automatically recognize emotions in the Italian language, possibly also due to the lack of suitable emotional speech databases (EMOVO [19], COST2102 [20,21]). Nowadays, only two SER works have been done concerning the Italian language [22,23]. In particular, Mencattini et al. [23] extract 520 features from the pitch contour, energy, and amplitude modulation of the signal; then, a feature selection step is performed to reduce the space complexity. Gender detection is performed using Linear Discriminant Analysis (LDA) on a subset of 5 selected features used to implement two different emotion models, one for male and one for female speakers. The work achieved promising results, but the articulated features extraction and features selection procedure adopted results in a large computational effort that may not meet the requirements of the social robot. For this reason, it would be interesting to look for other solutions.

## 3. Materials and Methods

The architecture we propose is presented in Figure 2. Three modules have been included:SPEECH DETECTOR (SD) module: to detect and record the speaker utterance;GENDER RECOGNITION (GR) module: to recognize the speaker gender from the audio file generated in the previous step;SPEECH EMOTION RECOGNITION (SER) module: to decode the emotional state of the speaker.

**Figure 2 sensors-22-01714-f002:**
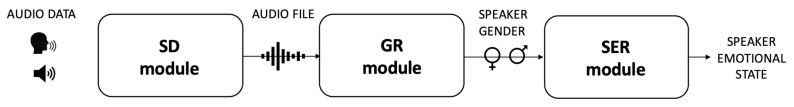
GESPER Architecture. Input audio data is sequentially processed by Speech Detector (SD), Gender Recognition (GR) and Speech Emotion Recognition (SER) modules for the final emotion decoding.

The final goal of the architecture is to decode the emotional state of the patient when he/she is talking. As already discussed, in this work we studied for an optimized GR classifier, thus we focused just on the first two modules of the architecture, in view of the future gender-specific SER implementation. The functional description of the Speech Detector and Gender Recognition module is provided in the following paragraphs.

The algorithm has been developed in the Python programming language. It is required for the software to be implemented on a lightweight and installable platform to be easily embedded on social robot hardware: to this aim, the algorithm was executed on a Raspberry Pi 4 Model B+ (1.5 GHz Quad-Core 64-bit ARM Cortex-A72 CPU—2.4/5 GHz WLAN ac—size: 9.6 × 7.2 × 3 cm—weight: 0.068 Kg) for testing the performance of the system as it is working in real-life settings. An omnidirectional condenser microphone, connected to the Raspberry Pi, was used as an acquisition device.

### 3.1. Speech Detector Module

The Speech Detector (SD) module was designed to detect speech input from the microphone and, at that point, save the audio data into an audio file. The SpeechRecognition open-source Python library (https://pypi.org/project/SpeechRecognition/, accessed on 20 January 2021) was used to this aim: detection is ensured by a continuous listening of the background environment that correctly discriminates speech from noise. The Google API of the library also provides a transcription of the utterance. For our study purposes, we currently do not use this information. However, in view of future works, it may be useful to keep track of what has been said, so as to search for keywords in the text and use them as additional information for the emotion recognition task.

Internet connection is required for the correct functioning of that library since connection and failure in speech understanding errors interrupt the program execution and no more audio data is recorded. To avoid this problem, both error types have been managed in our software to provide uninterrupted working. A detailed description of the SD module functioning is provided in Figure 3.

### 3.2. Gender Recognition Module

The GR module decides the speaker gender by processing the audio file generated in the SD module (Figure 4). A distinct Gaussian Mixture Model (GMM) was built for each gender; for the final decision, the resulting log-likelihoods were compared and the most likely gender is assigned to the speaker.

Training Database—Training of the models was performed on the Italian emotional database EMOVO [19] where 6 actors (3 males and 3 females) pronounced 14 different utterances (both sense and non-sense) in 7 emotional states (the “big six” disgust, fear, rage, joy, surprise, sadness and the neutral state), for a total of 588 audio signals. The use of this database allows us to train the models for a good distinction of gender, even in emotional conditions, where frequency components voice variations are typical.

Pre-processing—The generated audio was resampled at 22,050 kHz and, accordingly to the literature, they were framed with a window length of 0.03 s and an overlap of 0.015 s to ensure the stationarity condition required for the spectral features extraction [24].

A very simple Voice Activity Detection (VAD) was developed to distinguish speech and non-speech (or paused) frames from the spoken utterance recorded by the SD module. The energy was computed for each frame and compared to a threshold, set to 5% of the maximum audio data energy value. Frames with an energy higher than this threshold were classified as speech, while the others as non-speech. The energy threshold value was chosen empirically to ensure the correct classification of the segments.

Pseudo-code for the algorithm is shown in Algorithm 1. An example of the outcome of the VAD algorithm and selection of speech and non-speech frames is shown in Figure 5.

Noise reduction was performed by using an open-source Python library (https://pypi.org/project/noisereduce/, accessed on 9 February 2021)), which directly masks noise from the FFT of the audio data. The spectrum of the background noise was obtained from the non-speech frames of the utterance, After noise reduction, only speech frames were employed for features extraction.
**Algorithm 1** Voice Activity Detection (VAD)input← audio filerms← frame energy(input)threshold←0.05×max(rms)**for***i* in range(len(rms)) **do**   **if** rms[i]≥threshold **then**     frame *i* stored in speech   **else**     frame *i* stored in nonspeech   **end if****end for****return**speech,nonspeech

Features Extraction—For each frame, MFCC (numcep = 13, nfilt = 26) and Spectral Subband Centroids (SSCs) (nfilt = 13) were extracted using the open-source Python library python_speech_features (https://python-speech-features.readthedocs.io/en/latest/, accessed on 15 March 2021).

The SSCs features correspond to the centroid frequency of each subband. Popular areas of application of these features are speech recognition [25], speaker authentication, and voiceprint [26,27], where also a combination of SSCs with other features have proved to be more robust to noise [28]. In our work, SSC, MFCC, and their combination have been tested on both raw and pre-processed audio files (with VAD) for the choice of the best performance.

Model—Two Gaussian Mixture Models (GMM) were built: one was trained on female voices and the other on male voices. Both models were built with 4 components and full covariance. Parameters were chosen empirically by selecting the ones resulting in higher accuracy.

The gender of a speaker is obtained by comparing the log-likelihoods from both models.

## 4. Results

The previously described system has been evaluated by considering two aspects: the classifier performance of the GR module, and the speed of the software executed and tested on the Raspberry Pi platform in real-time settings.

### 4.1. Classifier Performance

The performance of the GR module as a classifier was first evaluated by using leave-one-speaker-out cross-validation on the EMOVO database. The overall best performance was achieved with SSCs features (nfilt = 13) alone (97.8% of accuracy) on the pre-processed audio file, as shown in Table 1. The accuracy value obtained for the gender distinction is comparable to those reported in the literature, as shown in Table 2.

For further testing, an external evaluation database of 3 male and 3 female speakers (aged between 25 and 5) was created to simulate a real-life scenario. For each speaker, 10 audio segments (lasting from 3 to 8 s) were extracted manually from random YouTube videos recorded at different background noise and recording settings conditions. In this case, the resulting accuracy value is 96.7%. Unfortunately, it is not possible to make a comparison for this value. Most of the considered related works did not test the algorithm on a mismatched condition dataset, except for [18], where the achieved accuracy dropped to 94.7%. However, our classifier outperformed this result, proving to work accurately even in real conditions.

### 4.2. System Performance

The proposed architecture is expected to be implemented on a social robot for working in real-life environments. For the benefit of human-robot interaction timings, the developed modules should work quickly. Therefore, the processing speed of each module will be evaluated by simulating a real-time setting. The environment used for the test was a room with a background noise of −55 dBFS. A monitor speaker reproducing the external database audio segments (presented in Section 4.1) was placed 0.5 m far from the Raspberry Pi microphone. This arrangement emulates what and how a social robot would be hearing from a speaker in a real-life setting.

Results show that both for the SD and GR module the processing time depends on the utterance duration with a linear behavior (Figure 6); however, by comparing audio files of the same duration, it is evident that the GR module is not only faster than the SD, but also less variable. This is due to the fact that, unlike the GR module, the speed of the SD is also influenced by external factors such as the internet connection and the difficulty for the Google API to solve the transcription; all these reasons may contribute to the increase in processing time variability.

## 5. Discussion

The proposed system continuously listens to the environment providing the gender information of the speaker with high accuracy. However, the processing timings of the developed modules are widely variable depending on the duration and complexity of the pronounced utterances. This behavior contrasts the real-time application requirements.

A further current limitation that must be managed emerges in case of no connection for the SD module: in this condition, no distinction is made between unspoken/noise and spoken audio and the generated audio file is provided as input for the gender recognition in any case.

Finally, by the use of the SD and GR modules only, the system has not enough information to identify the speaker for which the emotional state has to be recognized. In fact, in a real-world application, the robot will be placed in the home, co-housing, or hospital environments where the patient to be monitored may not be the only speaker present, but there may be other individuals on whom the emotional recognition would be made indistinctly. In this scenario, it is evident the need to correctly identify the patient, also in a multi-speaker conversation.

## 6. Future Directions

In this section we present some ideas to face the above-mentioned limits, improving the performance of the current architecture, thus laying the foundations to tackle the second part of this two-level architecture.

### 6.1. Further Modules

The correct recognition of the patient to be monitored, even in a multi-speaker environment, requires the addition of two modules that are the Speaker Diarization and the Speaker Recognition ones.

Speaker Diarization module - It aims at identifying the user-specific speech segments in case of a multi-speaker conversation: it answers the question “who spoke when?” [29]. By the segmentation of a conversation in user-specific audio files, it is possible to consider only the audio file referred to a particular user of interest for the emotion recognition task.

Speaker Recognition module - It aims at recognizing the users of interest (for the emotional state knowledge or the simple recognition of a person) from the voice. The Speaker Recognition needs to be text-independent in order to recognize the speaker for every pronounced utterance.

### 6.2. Other Suggestions

Modules Parallelisation - As seen in the Results section, the large variability of the modules processing times may be an issue for the performance of the system in real-time. A parallelisation of the modules is required to provide a reduction of the timings.

Database extension - The EMOVO database used for the training consists of only six actors: its limitation does not allow to take into account the real-world inter-speaker variability, both for gender and for the emotion recognition task. An enrichment of that database or the development of a new one is necessary to obtain more significant results for both tasks.

## 7. Conclusions

In this work, we presented the idea of a Speech Emotion Recognition architecture two-level based, aimed to be equipped on a social robot. We focused on the first level of the architecture, in particular on the Speech Detector and Gender Recognition modules. We achieved a recognition accuracy value close to 98%, comparable to the state-of-art related works. Our results proved a good classification performance also in a real-life scenario with varying environmental conditions.

However, the variability of the software processing speed represents a current limit to be managed. At the same time, the proposed architecture does not allow the recognition of the specific user of interest whose emotional state wants to be known. In view of future works, it is necessary to implement additional modules in the architecture and solve the limitations found so far.

## Figures and Tables

**Figure 1 sensors-22-01714-f001:**
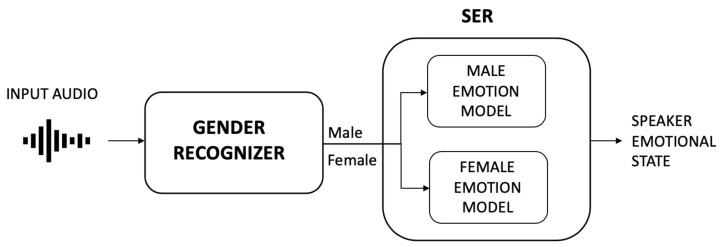
Example of a two-level recognizer architecture for Speech Emotion Recognition (SER).

**Figure 3 sensors-22-01714-f003:**
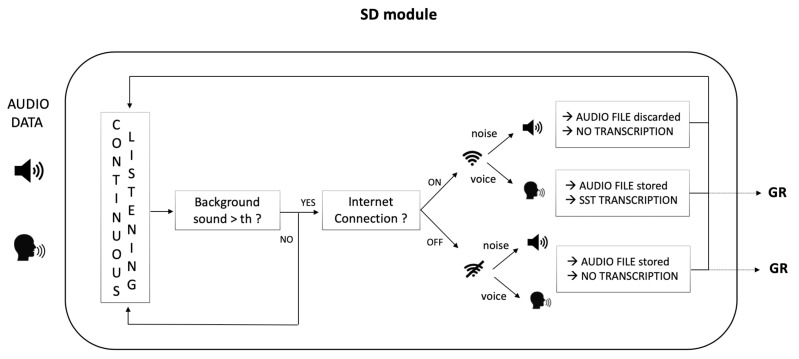
Speech Detector (SD) module functioning. The module continuously listens to the environment through the microphone and starts to record when an energy value above a certain threshold (dynamically set on the basis of the background noise, to avoid noise recording) is detected. Two possible cases can occur: INTERNET CONNECTION ON. If the user is speaking: recording and transcription occur; even if the user is not speaking Italian (e.g., another language or dialect) the Google API tries to fit a transcription anyway (but it might result in erroneous results). If noises are detected—mostly referring to those characterizing hospital/house environments such as medical instrumentation, lane noises, TV background, mumbling, as well as any kind of sound different from speech—no audio is recorded and no transcription occurs. INTERNET CONNECTION OFF. The module is not able to distinguish between voice and noise and, in both cases, audio is recorded and processed indiscriminately in subsequent blocks. No transcription occurs.

**Figure 4 sensors-22-01714-f004:**
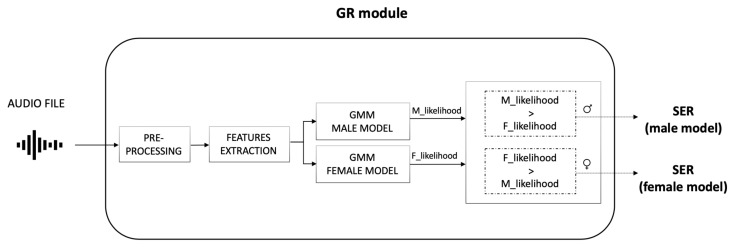
Gender Recognition (GR) module functioning. The module takes the audio file as input. After a pre-processing step, features are extracted and evaluated using the two GMM models (one for male and one for female speakers). The resulting log-likelihoods are then compared for the final decision.

**Figure 5 sensors-22-01714-f005:**
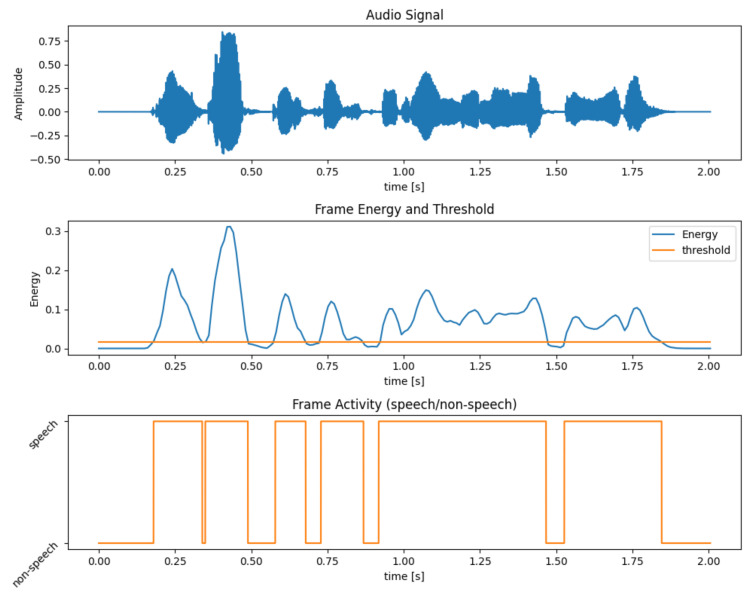
Example of Voice Activity Detection (VAD) algorithm functioning. On the top, the audio signal. On the center, the energy of the signal and the selected threshold (5% of maximum energy). On the bottom, the VAD output, speech and non-speech frames.

**Figure 6 sensors-22-01714-f006:**
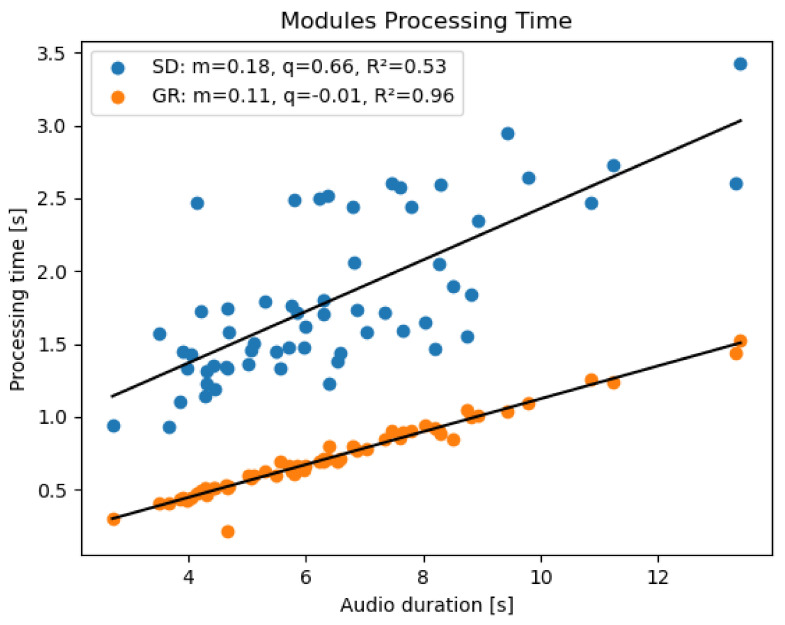
Processing time of the Speech Detector (blue) and Gender Recognition (orange) modules executed on Raspberry Pi in a real-time scenario. For each module, the slope, intercept and coefficient of determination (m, q, and R${}^{2}$ respectively) are displayed.

**Table 1 sensors-22-01714-t001:** Cross-validation accuracy of the Gender Recognition module using SSC, MFCC and their combination on raw/pre-processed audio file.

Features	NO VAD	VAD
SSC	89.8%	97.8%
MFCC	53.4%	90.6%
MFCC + SSC	72.4%	94.0%

**Table 2 sensors-22-01714-t002:** Comparison between Gender Recognition accuracy of GESPER and literature related works.

Related Work	Accuracy
Bisio et al. [15]	100.0%
Vinay et al. [11]	100.0%
Kabil et al [18]	99.8%
Alkhawaldeh [17]	99.7%
Shaqra et al. [12]	99.6%
Gesper	97.8%
Vogt et al. [10]	91.8%
Ramdinmawii et al. [16]	69.2%

## Data Availability

Not applicable.

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
