# Peer review of "Gender Identification in a Two-Level Hierarchical Speech Emotion Recognition System for an Italian Social Robot"

_sensors, 2022, doi:10.3390/s22051714_

Round 1
Reviewer 1 Report
The paper is well-written and touches on two important aspects of research: the analysis of elderly people speech and the running speed of the developed models. However, before the paper can be published, the below-mentioned problems need to be addressed.
1) The focus of the study is gender recognition, however, the authors suggest that they propose a "... Speech Emotion Recognition (SER) system". The title and the content of the article should be re-considered in order not to confuse the readers what the article is about. It is about gender recognition, not emotion recognition.
2) Authors state that "The system was designed to be installed on social robots for hospitalised or living at home elderly patients monitoring". However, the database that authors use do not feature elderly speech, and therefore, can not be used to assess the performance on elderly people. It is well-known that elderly speech is significantly different from adults due to changes to vocal tract that come with age.
3) Section 4.1 provides insufficient description about the collected data. What was the criterion for choosing the data? Did the authors select youtube videos manually or randomly? How long were the audio files? How old were the speakers? Why didn’t the authors test the system on elderly speech, if this is the final goal of the system? The classifier performance evaluation should be considered incorrect due to unclear data collection process.
4) The difference between the VAD and SD modules is confusing. According to authors, SD module was designed to detect speech input from the microphone, and VAD was developed in GR module to distinguish voiced and unvoiced segments. But how can the speech input be unvoiced?
5) Both voiced and unvoiced segments were passed through nose reduction? What happens to unvoiced segments then? Authors state that "The voiced frames obtained after noise reduction were employed for features extraction." Nose reduction usually affects the quality of voice and may significantly influence the emotion recognition stage. So it's not clear if this data preprocessing is actually suitable for emotion recognition module.
6) Figure 5 indicates that the processing times of SD module are far too long to be considered suitable for real-time systems. Even short audio files take more than 1 second to process. Since the system is continuously listening, this running time is unacceptable for real-time implementation. Parallelization does not seem possible due to consecutive connection of the modules (the output of SD is used in GR). On the other hand, if the system is continuously listening, and is only focused on one particular patient, why is it necessary to recognize his gender for every new utterance? I think it is possible to re-consider the whole application scenario for the developed system and make it more optimized.
Author Response
1) The focus of the study is gender recognition, however, the authors suggest that they propose a "... Speech Emotion Recognition (SER) system". The title and the content of the article should be re-considered in order not to confuse the readers what the article is about. It is about gender recognition, not emotion recognition.
The final goal of our project is Speech Emotion Recognition. However, as we stated in the article, the attention for this work is focused on the optimisation of the first level of the proposed architecture, i.e. Gender Recognition. Thus, we confirm that the article’s main subject is Gender Recognition, used for the final Speech Emotion Recognition.
2) Authors state that "The system was designed to be installed on social robots for hospitalised or living at home elderly patients monitoring". However, the database that authors use do not feature elderly speech, and therefore, can not be used to assess the performance on elderly people. It is well-known that elderly speech is significantly different from adults due to changes to vocal tract that come with age.
Since Emotion Recognition strictly depends on different aspects, such as social context, we decided to use only italian speakers for training the classifier. The only available emotional italian database is the one we used, EMOVO (speakers age ranging between 23 and 30). This is the reason why we tested on a similar age range in real-time.
3) Section 4.1 provides insufficient description about the collected data. What was the criterion for choosing the data? Did the authors select youtube videos manually or randomly? How long were the audio files? How old were the speakers? Why didn’t the authors test the system on elderly speech, if this is the final goal of the system? The classifier performance evaluation should be considered incorrect due to unclear data collection process.
Appropriate changes to the text were made in section 3.2 and 4.1 in order to better clarify the unclear contents.
4) The difference between the VAD and SD modules is confusing. According to authors, SD module was designed to detect speech input from the microphone, and VAD was developed in GR module to distinguish voiced and unvoiced segments. But how can the speech input be unvoiced?
Appropriate changes to the text were made in section 3.2 and 4.1 in order to better clarify the unclear contents.
5) Both voiced and unvoiced segments were passed through noise reduction? What happens to unvoiced segments then? Authors state that "The voiced frames obtained after noise reduction were employed for features extraction." Noise reduction usually affects the quality of voice and may significantly influence the emotion recognition stage. So it's not clear if this data preprocessing is actually suitable for emotion recognition module.
We agree that noise reduction may influence Emotion Recognition. However, here we applied it just for the GR module. This does not imply that it will be used also for SER. The gender information is exploited by the SER module only to know which emotion model to use, but the pre-processing stage is independent and could be different for each module.
6) Figure 5 indicates that the processing times of SD module are far too long to be considered suitable for real-time systems. Even short audio files take more than 1 second to process. Since the system is continuously listening, this running time is unacceptable for real-time implementation. Parallelisation does not seem possible due to consecutive connection of the modules (the output of SD is used in GR). On the other hand, if the system is continuously listening, and is only focused on one particular patient, why is it necessary to recognise his gender for every new utterance? I think it is possible to re-consider the whole application scenario for the developed system and make it more optimised.
Despite the few seconds of processing time needed, both for the SD and GR module, the system continuously listens to the environment, even during processing, without losing data. For this reason, the system is suitable for real-time applications.
Gender Recognition is needed at every utterance because the social robot and the module may work also in environments where more speaker or patients are present.
Reviewer 2 Report
Quite a superficial (missing basic technical information) survey of existing methods for speech preprocessing, i.e. speech preprocessing, feature extractions, feature selection and fusion, classification techniques - feature modeling, especially focused on gender recognition (GR) task.
Novelty and contribution are vague as comparable results with other approaches were achieved using standard methods, and generality is limited only to Italian language.
Some sentences are difficult to understand due to non-standard grammar constructions.
Using a speech recognition system just to detect speech segments for GR via Google API is a rather computational (resources) expensive solution that is moreover depended on other platforms and internet connections.
Having 3 males and 3 females or 6 additional speakers – using extended dataset (even though not clearly explained how these datasets were mutually used) is utterly insufficient to train and judge the performance of a gender recognition system.
There are 3 different terms used and apparently mixed up, i.e. VAD, SD, voiced / unvoiced. Even though they may seem similar they are different and this should be clarified what exactly was meant in every processing stage. Usually voiced and unvoiced in speech processing denotes parts, where fundamental frequency in speech is present or not, that differs from presence or absence of speech itself (VAD).
Critical details regarding both speech processing and feature modeling are missing, e.g. frequency ranges, number of filter banks (MFCC), number of MFCC, number of GMM mixtures, how they were trained/initialized, what were the lengths of speech blocks upon which the classification was performed (it is very important for an accuracy), etc. Moreover were the used settings somehow optimized?
To judge performance of the system in mismatch scenarios more information is needed as: SNRs, amount and type of reverberation, types of noises (both additional and convolutional).
Why just MFCC and SSC features were tested, there are so many of them (even hundreds), e.g. basic, derived, acoustic, prosodic, higher level, time- frequency domain, pre glottal, etc. Each group is suitable for particular application. Gender recognition is rather specific so there are other and maybe simpler and more effective features to test; except acoustic also prosodic (fundamental frequency).
The same holds for classification – modelling part, where SSC alone provided much better results than a combination of MFCC and SSC. A proper classifier would probably eliminated this adverse / illogical phenomenon- outcome.
Better results were not provided in comparison to other standard methods that was commented / justified as the used methods are computationally less expensive. But no comparison to other methods in terms of computational load or processing times was provided. Thus this statement or justification is not supported by any evidence.
It was observed that SD is more time demanding than GR, which is almost always not true as GR task is at least one level higher than SD. It was naturally caused by using speech recognition system to detect speech, which is currently one of the most difficult task in speech processing. Here it is used “only” to detect speech, which seem to be wasting of resources.
Author Response
1) Quite a superficial (missing basic technical information) survey of existing methods for speech preprocessing, i.e. speech preprocessing, feature extractions, feature selection and fusion, classification techniques - feature modelling, especially focused on gender recognition (GR) task.
2) Novelty and contribution are vague as comparable results with other approaches were achieved using standard methods, and generality is limited only to Italian language.
3) Some sentences are difficult to understand due to non-standard grammar constructions.
4) Using a speech recognition system just to detect speech segments for GR via Google API is a rather computational (resources) expensive solution that is moreover depended on other platforms and internet connections.
We agree that the Speech Detector module uses an expensive computational effort, but its purpose is not only to detect speech for Gender Recognition. Transcription may be needed also for real-time sentiment analysis in the Emotion Recognition field.
5) Having 3 males and 3 females or 6 additional speakers – using extended dataset (even though not clearly explained how these datasets were mutually used) is utterly insufficient to train and judge the performance of a gender recognition system.
We decided to use only italian speakers for training the classifier, since the SER strictly depends on different aspects, such as social context and language. The only available emotional italian database is the one we used, EMOVO, with 3 male and 3 female speakers. We know that the dataset is not sufficient enough to achieve an accurate model and, as we stated in the article, we agree that the database needs to be extended.
6) There are 3 different terms used and apparently mixed up, i.e. VAD, SD, voiced / unvoiced. Even though they may seem similar they are different and this should be clarified what exactly was meant in every processing stage. Usually voiced and unvoiced in speech processing denotes parts, where fundamental frequency in speech is present or not, that differs from presence or absence of speech itself (VAD).
Appropriate changes to the text were made in section 3.2 and 4.1 in order to better clarify the unclear contents.
7) Critical details regarding both speech processing and feature modeling are missing, e.g. frequency ranges, number of filter banks (MFCC), number of MFCC, number of GMM mixtures, how they were trained/initialized, what were the lengths of speech blocks upon which the classification was performed (it is very important for an accuracy), etc. Moreover were the used settings somehow optimized?
Appropriate changes to the text were made in section 3.2 and 4.1 in order to better clarify the unclear contents.
8) To judge performance of the system in mismatch scenarios more information is needed as: SNRs, amount and type of reverberation, types of noises (both additional and convolutional).
9) Why just MFCC and SSC features were tested, there are so many of them (even hundreds), e.g. basic, derived, acoustic, prosodic, higher level, time- frequency domain, pre glottal, etc. Each group is suitable for particular application. Gender recognition is rather specific so there are other and maybe simpler and more effective features to test; except acoustic also prosodic (fundamental frequency).
We decided to consider a reduced number of features to ensure a low computational effort for the system. For this reason, we mainly tested MFCC since they are the most relevant features used in literature. We also considered SSC since they are used in literature as a complementary feature with MFCC for obtaining higher accuracy, especially in the field of Speaker and Speech Recognition.
10) The same holds for classification – modelling part, where SSC alone provided much better results than a combination of MFCC and SSC. A proper classifier would probably eliminated this adverse / illogical phenomenon- outcome.
11) Better results were not provided in comparison to other standard methods that was commented / justified as the used methods are computationally less expensive. But no comparison to other methods in terms of computational load or processing times was provided. Thus this statement or justification is not supported by any evidence.
12) It was observed that SD is more time demanding than GR, which is almost always not true as GR task is at least one level higher than SD. It was naturally caused by using speech recognition system to detect speech, which is currently one of the most difficult task in speech processing. Here it is used “only” to detect speech, which seem to be wasting of resources.
Please see point 4.
Reviewer 3 Report
The manuscript reports a study where a gender detection module has been tested in real conditions. The module is aimed for social robots and therefore it must run on modest hardware; a raspberry Py was used in the evaluation.
The results indicate that the recogniser works as expected. The gaussian mixture model-based solutions matches what is reported in literature. The processing also works reasonably efficiently but the chosen solution has somewhat random performance due to its network components. While some part of the novelty is that the work was done on Italian language material, there is no reasoning, why the language would make a difference.
Overall, the manuscript is fairly clear to read but I fail to see, what is the worthwhile contribution. I think the text should discuss more thoroughly what elements the audio processing must include, which of them are prone to errors and consume resources, and which parts are language dependent.
The current manuscript describes the evaluation done reasonably well but it seems to fail connecting the elements of the work done to theory and why different things are done. Now, the different elements and their meaning are left for the reader to make assumptions on. As a concrete example, if the authors could report how much the later modules in processing can benefit from the gender recognition, that could be interesting and valuable. However, as the actual emotion models are not apparently available yet, this is perhaps impossible. How ever, at least it could be discussed.
Author Response
Overall, the manuscript is fairly clear to read but I fail to see, what is the worthwhile contribution.
We developed a woking sensorial node dedicated to analyse human voices in order to assess speaker's emotional state. Our contribution must be viewed as the realisation of a real system inside a real robot. We are still working on this project in order to improve its performances for better results.
As a concrete example, if the authors could report how much the later modules in processing can benefit from the gender recognition, that could be interesting and valuable.
Since the robot can be used in environments where more speakers are present we need to identify who is speaking. The real scenario can be quite challenging and we want something that really works.
Round 2
Reviewer 1 Report
Among the 6 questions raised, the authors provided satisfactory answers and clarifications only for the Questions 2 and 3. Below is the additional discussion for the previously raised questions, as well as some additional comments at the end.
Question 1. I believe you can not state that you propose a “Speech Emotion Recognition (SER) system” (abstract line 5) even if it is the final goal of your project. Since this particular article is focused on the Gender Recognition (GR) part, it is ok to say that the proposed GR system is designed to be a part of the SER system, but it is not ok to say that you propose a SER system and then only describe the GR part. One should be careful to make distinction between what you have done (the GR described in the article) and what you intend to do in the future (the whole SER project). For now, you just promise that you will implement SER in the future. But what if you don’t? The content of the given article should still remain valid regardless whether or not the future SER is implemented.
Please try to think from a reader’s perspective. If I am interested in GR performance, is this article going to be useful for me? Probably, yes. But if I am interested in SER performance, is this article going to be useful for me? Probably, not.
For this reason, the related works section is also confusing. In this given article, are you addressing the GR problem, or the SER problem? If your focus is the GR, then the background on SER methods is irrelevant to this given article, and you should focus on the GR-related works. Unfortunately, the background on GR is very limited. The references [18] and [19] are too old to be mentioned as the related works, and the [16] and [20] are also not up-to-date. There is no analysis of more recent works [21-23], how do these works compare to each other? What are their advantages and disadvantages? What is the state-of-the-art performance? What are the databases used? Why did the authors opt for GMM models and not one of the mentioned CNNs? The Spectral Subband Centroids (SSCs) features used by the authors and corresponding references [25-27] should also be mentioned in this section.
Question 4. There is still confusion between the SD and VAD modules. They seem to do the same thing – energy-based threshold. Why is it necessary to perform energy-based thresholding 2 times? If the VAD module distinguishes speech and non-speech audio segments when the energy is above a certain threshold, why is the SD module necessary, if it performs the same thing (starts to record when an energy value is above a certain threshold)? Figure 5 is actually redundant as it shows a typical operation of energy-based VAD. What is necessary is to show the difference in operation between the SD and VAD, as they seem to do the same thing.
Also, the authors state that the VAD threshold is set to 5% of the maximum audio data energy value (line 160). But then, the authors state that the energy threshold value was chosen empirically (line 162). Which one is correct?
Question 5. According to the author’s description of the preprocessing steps, the Figure 2 shows an incorrect pipeline. Figure 2 implies a sequential data processing, from one module to another. However, if each module has its own preprocessing steps, then there should be additional data flow channels from the audio data to each module, i.e. the audio data should be able to by-pass the GR module to reach the SER module intact.
Question 6. It is necessary to clarify, what do the authors mean by “real-time”? What are the requirements for the real-time systems? Often, the requirements are stated as the maximum response time, which is usually defined in a range of milliseconds for frame-based processing. For utterance-based processing, the requirement is usually defined as the processing time being less than the recording time. Here, an important question that needs to be answered is: what is the expected length of an audio time during system operation? If a speaker is continuously speaking, how does the system know, when is the start of an utterance, and when is the end?
Another concern regarding the real-time operation of the system is the noise masking from the FFT of the audio. The authors state that the noise spectrum is obtained from non-speech frames. It is not clear if these non-speech frames come after the SD module or the VAD module. If they come after the SD module, then it will not be possible to obtain non-speech frames without internet connection, as SD module can not distinguish speech from non-speech. Now, from the author’s description it is also not clear if the system is meant to operate in an online mode, offline mode, or both? Is the GR module optimized for both noise-reduced and full noise recordings?
Additional comments.
- It is not clear how do authors define the “computational effort”? How do they compare the “computational effort” of 2 models, and how do they try to reduce it? Is it in the number of computational operations, the number of trainable parameters, the speed of training, or the speed of a single prediction? Please note that in abstract line 10 the sentence “Hence, the importance of reducing the software computational effort of the architecture also minimizing the hardware bulkiness, in order for the system to be suitable for social robots.” is incomplete and therefore the meaning is not clear (do the authors simply stress the importance or they specifically target certain goals?). Line 56 states that the “the computational effort of the software has been reduced to the minimum”; what specifically authors have done to reduce the “computational effort”, and how does the achieved “computational effort” compare to other models? These are important questions to evaluate the proposed system.
- In the related works section, some references are missing. For example, the first sentence (line 60) says “Most of the state-of-art SER studies implement …”, but no references are given, which SOTA studies are meant? Later in the paragraph the authors cited various works that do not reflect the current SOTA. In particular, reference 17 may not be used as SOTA for RAVDESS, as compared, for example, with recent works “A CNN-Assisted Enhanced Audio Signal Processing for Speech Emotion Recognition” by Kwon et. al.
- Minor grammar improvements:
Line 164 – the word “Figure” is missing before 5.
Line 219 – “the proposed system continuously listen” should be changed to “the proposed system continuously listens” or “the proposed systems continuously listen”
Overall, my advice to the authors is to be more careful with the words you use, as some of the important phrases are used too liberal without any justification (for example, some works are called state-of-the-art, although they are not state-of-the-art; authors state that they reduce computational effort but they dont mention how; authors state that they propose a SER system although they only proposed a GR system, and so on).
Good luck on improving the article!
Author Response
Question 1. I believe you can not state that you propose a “Speech Emotion Recognition (SER) system” (abstract line 5) even if it is the final goal of your project. Since this particular article is focused on the Gender Recognition (GR) part, it is ok to say that the proposed GR system is designed to be a part of the SER system, but it is not ok to say that you propose a SER system and then only describe the GR part. One should be careful to make distinction between what you have done (the GR described in the article) and what you intend to do in the future (the whole SER project). For now, you just promise that you will implement SER in the future. But what if you don’t? The content of the given article should still remain valid regardless whether or not the future SER is implemented.
Please try to think from a reader’s perspective. If I am interested in GR performance, is this article going to be useful for me? Probably, yes. But if I am interested in SER performance, is this article going to be useful for me? Probably, not.
For this reason, the related works section is also confusing. In this given article, are you addressing the GR problem, or the SER problem? If your focus is the GR, then the background on SER methods is irrelevant to this given article, and you should focus on the GR-related works. Unfortunately, the background on GR is very limited. The references [18] and [19] are too old to be mentioned as the related works, and the [16] and [20] are also not up-to-date. There is no analysis of more recent works [21-23], how do these works compare to each other? What are their advantages and disadvantages? What is the state-of-the-art performance? What are the databases used? Why did the authors opt for GMM models and not one of the mentioned CNNs? The Spectral Subband Centroids (SSCs) features used by the authors and corresponding references [25-27] should also be mentioned in this section.
We agree that some sentences in the paper may mislead the readers in thinking the article is about Speech Emotion Recognition. We rewrote some parts in the text in order to better clarify the aim of the article.
About the Related Works section, please keep in mind that the purpose of the paper is not to implement a generic Gender Recognition algorithm, but a GR module FOR Speech Emotion Recognition. SER is strictly connected to GR in this article, and most of the suggestions in this work (the architecture of the system, the used databases, the algorithms, etc.) were made specifically for providing a Gender Recognition module that would perform well for a SER system. For this reasons, in the Related Works section we opted for mainly referencing those works providing a Gender Recognition module for a Speech Emotion Recognition system.
No details have been added about the databases and methods used in this works as these comparisons are out of scope of this paper, but you can still reach this information from the same references.
The references [25-27] have not been added in this paragraph because they are not attributable to the "related works" section. The discovery of these features comes from works carried out in parallel with this study, that led us to experiment sscs in this area, as in any case they are characteristic features of the voice and much used in terms of speech recognition.
Question 4. There is still confusion between the SD and VAD modules. They seem to do the same thing – energy-based threshold. Why is it necessary to perform energy-based thresholding 2 times? If the VAD module distinguishes speech and non-speech audio segments when the energy is above a certain threshold, why is the SD module necessary, if it performs the same thing (starts to record when an energy value is above a certain threshold)? Figure 5 is actually redundant as it shows a typical operation of energy-based VAD. What is necessary is to show the difference in operation between the SD and VAD, as they seem to do the same thing.
Also, the authors state that the VAD threshold is set to 5% of the maximum audio data energy value (line 160). But then, the authors state that the energy threshold value was chosen empirically (line 162). Which one is correct?
Yes, both SD and VAD are energy-based threshold. But the task of SD is to understand when a person is speaking, from the beginning to the end of an utterance: the energy threshold defines the starting point, while the end is automatically defined after 0.5 s of silence with respect to the last "sound" detected. In this way we guarantee to record the whole utterance from the speaker. Then, the VAD is used as a pre-processing step to remove any frame (30 ms) of apparent “silence” within the utterance, in order to process just the spoken frames and not the ones that could, for instance, be unvoiced due to the pause between one word and the other.Question 5. According to the author’s description of the preprocessing steps, the Figure 2 shows an incorrect pipeline. Figure 2 implies a sequential data processing, from one module to another. However, if each module has its own preprocessing steps, then there should be additional data flow channels from the audio data to each module, i.e. the audio data should be able to by-pass the GR module to reach the SER module intact.
The energy threshold chosen empirically is indeed the 5% on the maximum audio data energy value. We tried other values for this parameter and 5% was the one that performed best.
Question 5. According to the author’s description of the preprocessing steps, the Figure 2 shows an incorrect pipeline. Figure 2 implies a sequential data processing, from one module to another. However, if each module has its own preprocessing steps, then there should be additional data flow channels from the audio data to each module, i.e. the audio data should be able to by-pass the GR module to reach the SER module intact.
Figure 2 has the scope to solely show the different steps of the investigated architecture level, and these steps are indeed sequentials. The arrows simply show a "passing" of information from one module to another, but what will be passed from the GR to SER module has not been studied yet. Investigation on additional data flows is reserved for the future
Question 6. It is necessary to clarify, what do the authors mean by “real-time”? What are the requirements for the real-time systems? Often, the requirements are stated as the maximum response time, which is usually defined in a range of milliseconds for frame-based processing. For utterance-based processing, the requirement is usually defined as the processing time being less than the recording time. Here, an important question that needs to be answered is: what is the expected length of an audio time during system operation? If a speaker is continuously speaking, how does the system know, when is the start of an utterance, and when is the end?
Another concern regarding the real-time operation of the system is the noise masking from the FFT of the audio. The authors state that the noise spectrum is obtained from non-speech frames. It is not clear if these non-speech frames come after the SD module or the VAD module. If they come after the SD module, then it will not be possible to obtain non-speech frames without internet connection, as SD module can not distinguish speech from non-speech. Now, from the author’s description it is also not clear if the system is meant to operate in an online mode, offline mode, or both? Is the GR module optimized for both noise-reduced and full noise recordings?
The final goal of the system is to provide a Speech Emotion Recognition module for a social robot, so what is important to achieve is for the Speech Detector to catch every spoken utterance without losing any relevent information. We tested the system and proved that the system could sustain a simple conversation with a subject, in which sentences are usually below 10 seconds, which is the intended use of the robot. In this sense, we can safely state that the system and the robot work in a real-time environment.
As already stated in the answer to question 4, SD module provides the whole utterance, while VAD works on a 30 ms frame level to extract speech and non-speech frames. Noise spectrum is obtained from non-speech frames, defined by VAD pre-processing. About the SD module, as specified in the text, "Internet connection is required for a correct functioning of that library", so we can confirm that the real-time system has been thought to work in an enviromnent with a stable Internet connection. However, in case of occasional network failures, we ensure to still record something (that can be noise, voice or both), losing just the trascription functioning of the library. The obtained audio data undergoes all the other subsequent steps of the system, so it is pre-processed with VAD and noise-reduction before features extraction and gender detection. Obviusly, if there is no internet connection, in case of high level of ambient noise, there is the possibility for the SD to detect a non-speech segment and that the audio data, which does not contain a spoken utterance, undergoes all the processing steps. However, internet connection failure are supposed to occur very rarely in the working environment of the social robot, hence not affecting the overall performance of the system.
Additional comments.
- It is not clear how do authors define the “computational effort”? How do they compare the “computational effort” of 2 models, and how do they try to reduce it? Is it in the number of computational operations, the number of trainable parameters, the speed of training, or the speed of a single prediction? Please note that in abstract line 10 the sentence “Hence, the importance of reducing the software computational effort of the architecture also minimizing the hardware bulkiness, in order for the system to be suitable for social robots.” is incomplete and therefore the meaning is not clear (do the authors simply stress the importance or they specifically target certain goals?). Line 56 states that the “the computational effort of the software has been reduced to the minimum”; what specifically authors have done to reduce the “computational effort”, and how does the achieved “computational effort” compare to other models? These are important questions to evaluate the proposed system.
The computational effort has to be intended mainly from the algorithm complexity point of view by reducing the number of features and the complexity of the preprocessing step. In particular, in this work we found the minimum number of features (13 SSC) as a trade-off between performance and complexity of the developed algorithm.
- In the related works section, some references are missing. For example, the first sentence (line 60) says “Most of the state-of-art SER studies implement …”, but no references are given, which SOTA studies are meant? Later in the paragraph the authors cited various works that do not reflect the current SOTA. In particular, reference 17 may not be used as SOTA for RAVDESS, as compared, for example, with recent works “A CNN-Assisted Enhanced Audio Signal Processing for Speech Emotion Recognition” by Kwon et. al.
With SOTA studies we meant those works explained later in the same paragraph. For clarity we explicity added the references at the beginning.
- Minor grammar improvements:
Line 164 – the word “Figure” is missing before 5.
Line 219 – “the proposed system continuously listen” should be changed to “the proposed system continuously listens” or “the proposed systems continuously listen”
Done.
Overall, my advice to the authors is to be more careful with the words you use, as some of the important phrases are used too liberal without any justification (for example, some works are called state-of-the-art, although they are not state-of-the-art; authors state that they reduce computational effort but they dont mention how; authors state that they propose a SER system although they only proposed a GR system, and so on).
Thank you for your suggestions, we hope we clarified any doubt.
Reviewer 2 Report
In the original review there were 12 comments and only 6 were party addressed, at least commented. The 6 remaining have been left without any comments. Some of them are critical for the article/ evaluation/ comparison, namely 1), 10), and 11).
Author Response
In the original review there were 12 comments and only 6 were party addressed, at least commented. The 6 remaining have been left without any comments. Some of them are critical for the article/ evaluation/ comparison, namely 1), 10), and 11).
Question 1 - Quite a superficial (missing basic technical information) survey of existing methods for speech preprocessing, i.e. speech preprocessing, feature extractions, feature selection and fusion, classification techniques - feature modeling, especially focused on gender recognition (GR) task.
If you are referring to the Related Works section, please keep in mind that the purpose of the paper is not to implement a generic Gender Recognition algorithm, but a GR module for Speech Emotion Recognition. For these reasons, we opted for mainly referencing those works providing a Gender Recognition module for a SER system. No details have been added about the databases and methods used in this works as these comparisons are out of scope of this paper, but you can still reach this information from the same references.
Question 2 - Novelty and contribution are vague as comparable results with other approaches were achieved using standard methods, and generality is limited only to Italian language.
We agree that this work is limited to the italian language. However, the ambition of this study is to implement a Speech Emotion Recognition algorithm for an italian social robot. Currently, as we stated in the paper, only few works were done about (italian) language-specific SER, so we believe that our work is already a step forward. In particular, we implemented a sensorial node with suitable properties (lightweight, small, independent,…) for equipping a social robot.
Question 3 - Some sentences are difficult to understand due to non-standard grammar constructions.
Question 4 - Using a speech recognition system just to detect speech segments for GR via Google API is a rather computational (resources) expensive solution that is moreover depended on other platforms and internet connections.
We agree that the Speech Detector module uses an expensive computational effort, but its purpose is not only to detect speech for Gender Recognition. Transcription may be needed also for real-time sentiment analysis in the Emotion Recognition field.
Question 5 - Having 3 males and 3 females or 6 additional speakers – using extended dataset (even though not clearly explained how these datasets were mutually used) is utterly insufficient to train and judge the performance of a gender recognition system.
We decided to use only italian speakers for training the classifier, since the SER strictly depends on different aspects, such as social context and language. The only available emotional italian database is the one we used, EMOVO, with 3 male and 3 female speakers. We know that the dataset is not sufficient enough to achieve an accurate model and, as we stated in the article, we agree that the database needs to be extended.
Question 6 - There are 3 different terms used and apparently mixed up, i.e. VAD, SD, voiced / unvoiced. Even though they may seem similar they are different and this should be clarified what exactly was meant in every processing stage. Usually voiced and unvoiced in speech processing denotes parts, where fundamental frequency in speech is present or not, that differs from presence or absence of speech itself (VAD).
Appropriate changes to the text were made in section 3.2 and 4.1 in order to better clarify the unclear contents.
Question 7 - Critical details regarding both speech processing and feature modeling are missing, e.g. frequency ranges, number of filter banks (MFCC), number of MFCC, number of GMM mixtures, how they were trained/initialized, what were the lengths of speech blocks upon which the classification was performed (it is very important for an accuracy), etc. Moreover were the used settings somehow optimized?
Appropriate changes to the text were made in section 3.2 and 4.1 in order to better clarify the unclear contents.
Question 8 - To judge performance of the system in mismatch scenarios more information is needed as: SNRs, amount and type of reverberation, types of noises (both additional and convolutional).
Question 9 - Why just MFCC and SSC features were tested, there are so many of them (even hundreds), e.g. basic, derived, acoustic, prosodic, higher level, time- frequency domain, pre glottal, etc. Each group is suitable for particular application. Gender recognition is rather specific so there are other and maybe simpler and more effective features to test; except acoustic also prosodic (fundamental frequency).
We decided to consider a reduced number of features to ensure a low computational effort for the system. For this reason, we mainly tested MFCC since they are the most relevant features used in literature. We also considered SSC since they are used in literature as a complementary feature with MFCC for obtaining higher accuracy, especially in the field of Speaker and Speech Recognition.
Question 10 - The same holds for classification – modelling part, where SSC alone provided much better results than a combination of MFCC and SSC. A proper classifier would probably eliminated this adverse / illogical phenomenon- outcome.
The training database is very limited in both sample size and especially in the number of speakers. For this reason, a low number of features may prove to be more effective since it reduces overfitting on the training speakers.
Question 11 - Better results were not provided in comparison to other standard methods that was commented / justified as the used methods are computationally less expensive. But no comparison to other methods in terms of computational load or processing times was provided. Thus this statement or justification is not supported by any evidence.
We did not intend to state that our method is better than the others, but that we reduced the complexity of the algorithm to keep it suitable for a lightweight platform. The computational effort has to be intended mainly from the algorithm complexity point of view by reducing the number of features and the complexity of the preprocessing step. In particular, in this work we found the minimum number of features (13 SSC) as a trade-off between performance and complexity of the developed algorithm.
Question 12 - It was observed that SD is more time demanding than GR, which is almost always not true as GR task is at least one level higher than SD. It was naturally caused by using speech recognition system to detect speech, which is currently one of the most difficult task in speech processing. Here it is used “only” to detect speech, which seem to be wasting of resources.
See point 4.
Reviewer 3 Report
The manuscript reports work on hierarchical approach to emotion recognition. While novelty is limited and the problem space has not been thoroughly covered, the work is well written and easy to follow.
Author Response
The manuscript reports work on hierarchical approach to emotion recognition. While novelty is limited and the problem space has not been thoroughly covered, the work is well written and easy to follow.
Thank you for your comments.